GEDTM30: global ensemble digital terrain model at 30 m and derived multiscale terrain variables

Ho Yu-Feng 1 yu-feng.ho@opengeohub.org
http://orcid.org/0000-0001-5073-5572 Grohmann Carlos H. 2
Lindsay John 3
http://orcid.org/0000-0001-6336-7801 Reuter Hannes I. 4 5
http://orcid.org/0000-0003-1589-0467 Parente Leandro 1
http://orcid.org/0000-0002-0962-6478 Witjes Martijn 1
http://orcid.org/0000-0002-9921-5129 Hengl Tomislav 1
1 OpenGeoHub , Doorwerth, Gelderland , Netherlands
2 Institute of Astronomy, Geophysics and Atmospheric Sciences, Universidade de São Paulo , São Paulo , Brazil
3 Department of Geography, Environment & Geomatic, University of Guelph , Guelph , Canada
4 Eurostat, European Commission , Luxembourg , Luxembourg
5 Gisxperts , Trier , Germany
Cavalli Marco
Electronic publication date: 2025 Jul 24
Publication date: 2025
Volume: 13
Electronic Location ID: e19673
Received 2025 Mar 21; Accepted 2025 Jun 9
Copyright: © 2025 Ho et al.
Copyright year: 2025
Copyright holder: Ho et al.
License: This is an open access article distributed under the terms of the Creative Commons Attribution License, which permits unrestricted use, distribution, reproduction and adaptation in any medium and for any purpose provided that it is properly attributed. For attribution, the original author(s), title, publication source (PeerJ) and either DOI or URL of the article must be cited.
License URL: https://creativecommons.org/licenses/by/4.0/

Keywords: Digital elevation model, Digital terrain model, Hydrology, Topography, Geomorphometry, Open data, Data fusion, Machine learning, Transfer learning, Terrain variables

Funding: European Union’s Horizon Europe Research and Innovation Programme 101059548 Bezos Earth Fund Brazilian National Council for Scientific and Technological Development (CNPq) 311209/2021-1 São Paulo Research Foundation (FAPESP) 2023/11197-1 The Open-Earth-Monitor Cyberinfrastructure project has received funding from the European Union’s Horizon Europe research and innovation programme under grant agreement No. 101059548. The Global Pasture Watch under the Land Carbon Lab project has received funding from the Bezos Earth Fund. The work was supported by the Brazilian National Council for Scientific and Technological Development (CNPq) under Grant 311209/2021-1 and by the São Paulo Research Foundation (FAPESP) under Grant #2023/11197-1. There was no additional external funding received for this study. The funders had no role in study design, data collection and analysis, decision to publish, or preparation of the manuscript.

==============================
Production and validation of an open global ensemble digital terrain model (GEDTM30) and derived terrain variables at 1 arc-s spacing grid ( ∼30 m spatial resolution) is described. Copernicus DEM, ALOS World3D, and object height models were combined in a data fusion approach to generate a globally consistent digital terrain model (DTM). This DTM was then used to compute 15 standard terrain variables across six scales (30, 60, 120, 240, 480 and 960 m). A global-to-local transfer learning model framework with 5° × 5° tiling leveraged globally distributed lidar datasets: ICESat-2 ATL08 (best-fit terrain height) and GEDI02 (lowest mode elevation), totaling over 30 billion training points. A global model was initially fitted using ICESat-2 and GEDI, followed by locally optimized models per tile, ensuring both global consistency and local accuracy. Independent validation shows that GEDTM30 reduces Copernicus DEM RMSE by about 25.4% in built-up areas, 10.0% in regions with 10–50% tree cover, and 27.3% in areas with over 50% tree cover. Compared to state-of-the-art DTMs (MERIT DEM, FABDEM and FathomDEM), GEDTM30 achieves the lowest vertical errors when assessed with GNSS station records, yielding a standard deviation of 7.77 m, an RMSE of 10.69 m, and a mean error of 7.34 m. FathomDEM exhibited the lowest vertical RMSE when validated against independent reference DTMs. GEDTM30 was further used to generate multiscale variables of topography and hydrology through an optimized tiling workflow ( ∼800 tiles of 600 × 600 km with ∼16% overlap) based on the Equi7 grid system. The entire workflow was implemented in Python using GDAL and Whitebox Workflows. Visual inspection confirmed the absence of boundary artifacts and the preservation of hydrologic connectivity. The tiling-based implementation significantly reduces computational costs of generating large-scale DTMs and derived terrain variables. The GEDTM30 dataset and code are publicly available as Cloud-Optimized GeoTIFFs via Zenodo and the OpenLandMap STAC. Further fusion with local lidar-based DTMs and national DTMs is recommended to enhance local accuracy and level of detail.

Introduction

Topography or relief is best represented by digital elevation models (DEMs) (Guth et al., 2021; Hengl & Reuter, 2008). DEMs are often used to measure spatial variability in hydrological, geomorphological, and biological processes and are one of the fundamental layers of earth sciences, ecology, and biogeography. For example, runoff can be calculated directly from a DEM as a simplification of the underlying physics. The spatial distribution of plant species is associated with wetness and radiation indices. Hillshade, openness, and local topographic position have a significant impact on soil formation in soil type mapping. Topographic attributes are today widely used in the planning of data collection to facilitate hydrological monitoring, soil surveys, and biological surveys, for predictive soil and vegetation mapping and soil erosion modeling (Behrens et al., 2018; Maxwell & Shobe, 2022; Moore, Grayson & Ladson, 1991; Pike, Evans & Hengl, 2009). There is a growing demand for consistent and seamless global digital terrain models (DTMs) to support both regional and global applications. Although national and airborne-derived DEMs, such as the Actueel Hoogtebestand Nederland (AHN) in the Netherlands, DTM05 in Spain, and the 5 m DEM Grid of Australia, offer high resolution and accurate data, their coverage is limited to specific regions. Furthermore, these data sets often vary in resolution, projection, and data quality, which leads to inconsistencies in transboundary studies. Finally, many countries lack detailed terrain data or restrict their datasets from public access, further inhibiting wider global-scale analyses.

NASA’s Shuttle Radar Topography Mission (SRTM) DEM produced in 2000 has been one of the most widely used global environmental datasets produced directly by a space mission (Farr et al., 2007). At 5 m, Airbus WorldDEM (Neo) is possibly the most detailed terrain model with nominal vertical accuracy <4 m for 90% of the earth land surface; however, being a commercial product, it is beyond the budget of many research organizations. In recent years, several high-resolution (1 arc s, or 30 m at the Equator) global topographic datasets have been released as open data, including ASTER GDEM, ALOS AW3D30, NASADEM, TanDEM-X TDM90 and CopernicusDEM (Trevisani, Skrypitsyna & Florinsky, 2023). An up-to-date list of global DEMs is maintained on the OpenTopography portal (https://portal.opentopography.org/datasets?group=global). In the early 2000s, only 1–2 global high-resolution DEMs were widely accessible; today, at least three to four competing space missions are dedicated to mapping the world’s topography. This includes: ASTER GDEM, ALOS AW3D30 and CopernicusDEM. ASTER GDEM and AW3D30 are derived from optical imagery (photogrammetry), while SRTM, NASADEM, TanDEM-X and CopernicusDEM are based on radar interferometry (InSAR) (Simard et al., 2024). Although the TanDEM-X mission generated a 12 m global DEM, access to this data set is a commercial product of the German Space Center (DLR) and Airbus. Two reduced resolution DEMs, a 30 m and a 90 m version of TanDEM-X are available under the name “CopernicusDEM” (GLO30, GLO90) under an open data license (https://doi.org/10.5270/ESA-c5d3d65). AW3D30 and GLO30 can be considered state-of-the-art DEMs of the globe in 1 arc s grid spacing, as they have improved upon, and made partially redundant, their predecessors SRTM and ASTER GDEM both in level of vertical accuracy and in the accuracy of depiction of geomorphological features (Bielski et al., 2024; Guth et al., 2024).

Both CopernicusDEM and AW3D30 are digital surface models (DSMs), in the sense that they do not represent the true surface of the ground (except in open and unvegetated areas), but the top of the canopy and buildings or an intermediate elevation within the vegetation structure depending on the radar wavelength (C or X band in SRTM, CopernicusDEM) and the time of data acquisition due to loss of foliage in temperate forests (Guth & Geoffroy, 2021; Simard et al., 2024). Producing high-quality digital terrain models (DTMs) at a global scale is challenging. Light detection and ranging (LiDAR) is among the most accurate methods for estimating terrain height; however, its short wavelength constrains its applicability for DTM generation beyond the national scale. Additionally, the high cost of LiDAR technology makes it inaccessible for some economically disadvantaged countries, further hindering the feasibility of creating a consistent global DTM from solely LiDAR data. However, thanks to the development of spaceborne LiDAR such as ICESat or GEDI, one can not only evaluate the accuracy of global DEMs (Carabajal & Harding, 2005; Rodriguez, Morris & Belz, 2006), but also be used to model and remove the height of vegetation from densely forested areas (Robinson, Regetz & Guralnick, 2014; Simard et al., 2011). These advancements enable the extraction of bare-earth DTMs from DSM by filtering out above-ground objects. Yamazaki et al. (2017) was among the first to produce a canopy-corrected high resolution ( ∼3 arc s) global coverage DTM called Multi-Error-Removed Improved-Terrain (MERIT). MERIT was based on SRTM (v2.1) and ALOS AW3D30 (v1) and the removal of off-terrain objects involved bias corrections, noise filtering, and smoothing. As an approximation of the ground surface, it allowed global geomorphometry and hydrology mapping to be performed at high resolution (Amatulli et al., 2022, 2020). A significant improvement over MERIT is the Forest And Buildings removed Copernicus DEM (FABDEM), which is a 1 arc s grid spacing DTM (Hawker et al., 2022). In FABDEM, both buildings and vegetation were removed from the Copernicus DEM using a machine learning model, trained on airborne LiDAR. In addition, Dusseau, Zobel & Schwalm (2023), Kulp & Strauss (2019) and Pronk et al. (2024) produced the CoastalDEM, DiluviumDEM, and DeltaDTM also at 1 arc s, tackling coastal areas which requires higher accuracy (Meadows, Jones & Reinke, 2024). These are specialized DTMs that aim to serve modeling hydrological dynamics of sea level rise under the threat of climate change. The most recent global DTM is FathomDEM, which is probably the first global DTM in which object removal is based on the use of computer vision (Uhe et al., 2025).

With an increasing number of available sources of elevation data (global DEMs, orbital LiDAR sensors), as well as free and open-source software for geospatial data processing and machine learning, there is today a need for a fully open 1 arc s DTM of the world to support more complex Earth system modeling. Bielski et al. (2024) and Guth et al. (2024) provide comprehensive comparisons for the available global DEMs against LiDAR data, and conclude that GLO30 and AW3D30 are tied in quality but significantly more accurate than SRTM, NASADEM, and ASTER GDEM; FABDEM ranks best in terms of vertical accuracy compared to LiDAR DTMs. Although such comparison exercises might be a great interest to geomorphometry researchers, we think practically and believe that the majority of users require a single “best” ensemble reference estimate of world elevations, being less concerned with complexities such as pixel shift, noise, and/or artifacts attributable to various technological limitations, but primarily being focused on completeness, consistency, and ease of access and use. Ensemble DTMs can be produced using machine learning, although even more simple models or decision trees can be used locally. Machine learning-based models can be seen as “black box” since the code used to generate the DEM might not be publicly available, or if it is, an end user is unlikely (or expected) to possess the knowledge or hardware/software resources to replicate the processing pipeline (Bunge, 1963). For example, the ANADEM model (Laipelt et al., 2024) is available via the Google Earth Engine platform, along with a GitHub repository; however, we were unable to locate detailed documentation on how to use the code to replicate the data. More “simple” modeling approaches to derive an ensemble DTM often prove comparably accurate but are often fitted only locally. A global modeling would tend to possess bias of the hyperparameters if only data from a certain area were used. Topography, along with land cover, often has a high spatial autocorrelation and is dependent on geomorphology/lithology (Guo, Lenoir & Bonebrake, 2018; Tobler, 1970; Yu et al., 2024), so a global model trained with LiDAR data in temperate forests of Europe could not perform as well in tropical forests of Africa or South America. To mitigate this bias, a machine learning approach utilizing globally distributed sampling should be employed.

In addition to modeling DTM, the extraction of measures and spatial features from DEMs (geomorphometry) has been widely practiced and more variables have been derived to enrich the applications of DTM (Florinsky, 2017; Hengl & Reuter, 2008; Minár et al., 2024). However, deriving terrain variables using high-resolution global DTMs is cumbersome as there are many computational challenges (a single 1 arc s DEM in EPSG:4326 geographic coordinate system contains approximately 202 billion pixels globally). Amatulli et al. (2020) and (2022) pushed the boundaries of processing large global grids and produced terrain variables of the world to 90 m ( ∼3 arc s). Amatulli et al. (2020) specifically used the Equi7 grid (Bauer-Marschallinger, Sabel & Wagner, 2014), but also splitting the computation by watersheds (Amatulli et al., 2022) to divide the land mask into seamless parts and avoid artifacts (e.g., due to hydrological connectivity and the boundary effect). This research unveiled the potential to create high resolution, or even higher, such as 30 m ( ∼1 arc s) terrain variables of the world.

In this article, we present the Global Ensemble Digital Terrain Model (GEDTM30) as a fully open-source estimation of world topography at 1 arc s grid spacing based on multisource high-quality data and a global-to-local transfer learning approach. Grid spacing is interchangeable to (raster) resolution in this article. We have documented and distributed these data as Cloud Optimized GeoTIFFs, including all the necessary code to implement it and ensure reproducibility. In addition, the processing chain is hosted by OpenGeoHub, which allows GEDTM30 to be updated quickly whenever new data sources are released. We have also further optimized and operationalized the production of basic geomorphometric and hydrological variables at six standard grid spacing (30, 60, 120, 240, 480 and 960 m). The article is divided into two main parts: in the first part, we explain the construction of a globally consistent gap-free DTM using global distributed ICESat and GEDI points, and in the second part we describe how to efficiently derive, share, and use multiscale terrain variables from 30 m grid spacing up to 960 m. Portions of this text were previously published as part of a preprint (https://doi.org/10.21203/rs.3.rs-6280607/v1).

Materials and Methods

General framework

Our GEDTM30 modeling and mapping framework is illustrated in Fig. 1. It is based on two main parts: (1) 1 arc s global DTM generation from multisource data, and (2) multiscale terrain variables derivation. Global digital terrain modeling was based on Earth Observation data such as digital elevation models, multispectral satellite images, object modeling data (building/canopy height and extent), spaceborne global LiDAR data, and auxiliary data including coarser resolution DTM, landform classification, permanent ice cover and natural peaks from OpenStreeMap (OSM). To train the models, we used spaceborne LiDAR, GEDI and ICESat-2 to create reference terrain height. For our two models, we first trained a global model supported by stratified terrain samples from more than 30 billions samples, which was followed by a local model with localized samples adapted to the current processing extent.

Figure 1 The general processing workflows, key inputs and outputs for producing GEDTM30 and 15 terrain variables.

In the second part, we derived multiscale terrain variables from the produced global DTM. We first reprojected the produced DTM prepare tiles based on an extended Equi7 tiling system metric with grid spacing of 30 m (Bauer-Marschallinger, Sabel & Wagner, 2014). The extended Equi7 tiling system was applied with an overlap at the original tile border to avoid boundary effects and preserve hydrological connectivity. Each tile was then processed in parallel through the sequential derivation workflow. For the hydrological variables, a breaching and filling depression was applied prior to derivation. The processed tiles were subsequently then clipped back to the original extent and mosaicked to a continental extent in the Equi7 projection. Finally, continental maps were reprojected to EPSG:4326 to create a global mosaic with other irregular tiles. All outputs from this modeling, including reference samples, models and maps, are published on GitHub (https://github.com/openlandmap/GEDTM30), Zenodo platform (doi.org/10.5281/zenodo.14900180) and registered on OpenLandMap STAC (https://stac.openlandmap.org) under the permissive open access license CC-BY-4.0.

Input data

Training point data

As training points we consistently use GEDI and ICESat-2, two global exhaustive state-of-the-art sources of spaceborne LiDAR that also record terrain height (Dubayah et al., 2020; Markus et al., 2017). Global Ecosystem Dynamics Investigation (GEDI) is a full-waveform satellite LiDAR on the International Space Station, collecting 3D measurements over land (through recorded backscattered laser energy, i.e., waveforms) near-globally, covering from 51.6°N to 51.6°S and with a laser footprint diameter of about 25 m. To derive terrain height information, we use level 2 product, i.e., GEDI Level 2 Geolocated Elevation and Height Metrics products (GEDI02). The product provides waveform interpretation, including ground elevation, canopy top height, and relative height (RH) metrics. We first downloaded the GEDI data set covering the period between 2019.04.18 and 2023.03.16 from the Land Process Distributed Active Archive Center (LP DAAC). The original data set is stored in HDF5 format along GEDI’s orbital flight path. To facilitate processing, we restructured the data into 5° × 5° tile and applied a pre-filtering step. This filtering excluded low-quality GEDI points, i.e., GEDI shots with quality flag equal to 0, degradation flag larger than 0, and sensitivity smaller than 0.95. The final dataset comprised approximately five billion GEDI samples covering the latitudinal range of 51.6°N and 51.6°S.

The Ice, Cloud, and land Elevation Satellite-2 (ICESat-2) is the 2nd-generation of the laser altimeter ICESat mission, covering the Earth between 88°N and 88°S. ICESat-2 carries a photon-counting laser altimeter that initially aims to measure the elevation of ice sheets, glaciers, and sea ice, but can also be used to measure tree canopy height and land topography. For terrain modeling, we utilized the ATLAS/ICESat-2 L3A Land and Vegetation Height (ATL08) version 6 (Neuenschwander & Pitts, 2019). This data set provides height measurements along the track of both ground and canopy surfaces relative to the WGS84 ellipsoid, derived from the Advanced Topographic Laser Altimeter System (ATLAS). Initially, ATL08 processed canopy and ground elevations in fixed 100-m data segments; however, the release of version 6 in May 2023 introduced 20-m segments, offering improved resolution for terrain height (h_te_best_fit_20m) and canopy height (h_canopy_20m).

After data import, we combined ICESat-2 ATL08 and GEDI02 for terrain modeling to produce a complete consistent data set with the best estimates of terrain height. We specifically used the relative height at 95% (rh95) as the canopy height of GEDI02 in the counterparts to ATL08 canopy height (column h_canopy_20m) and terrain height (column elev_lowestmode) in the counterparts to ATL08 terrain height (h_te_best_fit_20m), despite the slight difference in shot diameter (20 m vs. 25 m). Both ICESat-2 ATL08 and GEDI02 represent elevations in the WGS84 vertical datum. In order to be compatible with existing digital elevation models, we applied a conversion from the WGS84 ellipsoid to the EGM2008 geoid as vertical reference for ICESat-2 ATL08 and GEDI02, using the 5 arc-degree resolution correction surface (Pavlis et al., 2012). Despite all these efforts to harmonize these data, we still observed discrepancies in terrain height between spaceborne LiDAR datasets and DSMs and many artifacts: for example, the presence of terrain samples over bodies of water, including lakes and oceans. Additionally, we noticed that the data set we prepared overrepresents bare-earth and underrepresents forest areas, which is suboptimal for object removal applications.

Cleaning and stratification of training data

To ensure a high-quality and representative training dataset, ICESat-2 and GEDI points must still be filtered and checked for anomalies. To remove any remaining anomalies, we overlaid the 5° × 5° tiled ICESat-2 and GEDI samples with GLO30, AW3D30, MERIT DEM, and JRC’s Global Surface Water (Pekel et al., 2016). The Global Surface Water dataset provides long-term water fraction measurements on a per-pixel basis from 1984 to 2021. We also excluded any samples in which the surface water fraction exceeded (>)20%. In addition, samples that showed significant discrepancies from DEMs were removed. Subsequently, for each tile, we computed the average standard deviation across all samples. The standard deviation for each sample was derived from four height values: the terrain height of the sample itself and the corresponding elevations from MERIT DEM, GLO30, and AW3D30. The mean of these standard deviations was used to establish a discrepancy threshold. Samples were removed if their absolute error, calculated as the difference between the MERIT DEM and the individual sample, exceeded two standard deviations. This method ensured that each tile had a reasonable discrepancy bound, reflective of the landform variability within the tile. To avoid excessively large discrepancy thresholds, we capped the bound at a maximum of 20 m in cases where the computed standard deviation was unusually high. After filtering, we randomly selected exactly 500,000 samples per tile to create a global sample pool. If the total number of filtered samples in a tile was below this threshold, all available samples were included. This approach ensured a balanced distribution of samples across global tiles before proceeding with stratification.

Following the initial filtering, the remaining samples from individual tiles were aggregated into a global dataset. To ensure a comprehensive representation of global terrain and aboveground objects, we implemented two stratification strategies: one for artificial objects and another for vegetation objects. Artificial objects include buildings, roads, and other man-made structures; Vegetation objects are forest, grassland, cropland, and also the green cover in urban parks, etc. The samples were initially separated using the binary mask that outlines the extent of human settlement (World Settlement Footprint, WSF2019) (Marconcini et al., 2021). The vegetation object samples were selected by locating pixels where WSF2019 is equal to 0, and the artificial object samples were selected by locating pixels where WSF2019 is equal to 1. Aftermath, we implemented the stratification on two sets of samples, respectively: Stratification of samples for vegetation objects removal: We stratified the vegetation object samples by canopy height in a 5 m interval until 30 m. There are seven strata, 0–5 m, 6–10 m, 11–15 m, 16–20 m, 20–25 m, 25–30 m, and above 30 m. The canopy height obtained from the LiDAR samples from h_canopy_20m or rh95 from ICESat-2 or GEDI respectively. In addition, to detect vegetation in various landforms, we further stratified the samples by Iwahashi & Pike’s (2007) landform classification. This layer is a 1 km spatial resolution raster data, presenting relief classes which are classified using an unsupervised nested-means algorithms and a three part geometric signature. The nested partitioning applies gradient, connectivity, and textures for three rounds from eight partitions to total 16 partitions, where half of the steeper land pixels feeds into a round of classification. In total, this two-tier stratification resulted in 102 nested strata (7 canopy height classes ×16 landform classes). We randomly selected approximately 45,000 samples per stratum, yielding a total of approximately five million samples for vegetation object removal.

Stratification of samples for artificial object removal: To remove artificial objects, samples were stratified according to building height using a 5 m interval up to 30 m, forming seven strata, 0–5 m, 6–10 m, 11–15 m, 16–20 m, 20–25 m, 25–30 m, and above 30 m. Building height data was extracted from the Global Human Settlement-Building Height dataset (GHS-BUILT-H R2022A) (Schiavina et al., 2022). This data set extracts the building height from the filtering of a composite of DEM and the filtering of satellite imagery using linear regression, providing global coverage at a 100-m resolution. For each building height stratum, we randomly selected 285,000 samples, ensuring a total of approximately two million samples for artificial-object removal.

During data processing, we identified a critical gap in the sample pool: the underrepresentation of mountain summits. These regions were either not measured by global LiDAR datasets or filtered out during the data cleaning process. Given the limitations of decision tree-based models, which extrapolate poorly beyond the observed data range, missing high-altitude samples could lead to inaccurate predictions in extreme elevation areas. To address this issue, we incorporated additional summit samples from OpenStreetMap (OSM) (OpenStreetMap Contributors, 2017) and the ICESat-2 and GEDI’s samples on the glaciers. The samples on glacier and permanent ice were identified using the University of Marylands Global Land Analysis & Discovery (GLAD)’s Global Land Cover and Land Use Change (GLCLUC) data set (Potapov et al., 2022), specifically selecting the points that are classified as permanent ice. Since altitude values in OpenStreetMap often differ from DEM-derived elevations, we replaced OSM summit heights by the average of the corresponding values from AW3D30 and GLO30 to ensure consistency with our model. Most of the OSM-derived summits were located in the Himalayas, the South Island of New Zealand, the Alps, and the Andes. Ultimately, we incorporated approximately 3,700 summit samples from OSM and randomly selected 50,000 additional LiDAR samples from glacier and ice-covered regions, enhancing the overall representativeness of high-altitude terrain and permanent ice of high latitude region in the dataset.

Covariates

To model global terrain height at a 1 arc-s grid spacing, we prepared an extensive set of covariates that capture essential information about terrain and surface objects. Two primary DSMs, GLO30 and AW3D30, were chosen as the state-of-the-art global open elevation data sets. However, both required preprocessing to ensure consistency in global modeling. For example, GLO30 provides no data for Azerbaijan, so we filled this gap using AW3D30 data. In contrast, AW3D30 contains sporadic but significant anomalies, including unrealistic craters and spikes, likely introduced during data processing. To correct for these inconsistencies, we derived the 95th percentile of absolute error between AW3D30 and GLO30 using the globally stratified sample set (as described in “Cleaning and stratification of training data”). This threshold was then used to identify and mask noisy pixels in AW3D30, replacing them with the corresponding values of GLO30. In addition, we incorporated global auxiliary data sets that significantly impact the removal of vegetation and artificial objects. Table 1 summarizes the covariates, including variable type, grid spacing, temporal range, and reference sources. The auxiliary layers can be roughly categorized into three thematic groups:

Vegetation object modeling: To estimate vegetation object height, we included models of tree cover and canopy height. Specifically, we selected the GLAD Global Tree Cover 2010 dataset, derived from annual Landsat 7 ETM+ composites spanning 2000–2012 (Hansen et al., 2013). This dataset provides a continuous canopy density measure ranging from 0 to 100 and has been widely used in machine learning-based digital terrain model (DTM) applications (Hawker et al., 2022; Hengl et al., 2020). For canopy height estimation, we incorporated two of the most detailed global models: the GLAD UMD Canopy Height Model (Potapov et al., 2021) and the ETH Canopy Height Model (Lang et al., 2023). The GLAD UMD model integrates multiple regional models through local fitting, while the ETH model employs an ensemble of convolutional neural networks (CNNs) trained under a sparse supervision framework. Despite both models being trained on GEDI LiDAR data, they exhibit systematic biases, where GLAD UMD tends to underestimate canopy height, whereas ETH tends to overestimate it. In addition to the canopy height model, we also applied sobel_filter with a 3 × 3 window from Whitebox Workflow (Lindsay, 2016) to detect the edge of canopy and added them as covariates.

Artificial object modeling: To model artificial objects, we included datasets that capture both the extent and height of built structures. The World Settlement Footprint 2019 (WSF2019) dataset was used to delineate human settlements. WSF2019, which also plays a role in sample stratification (“Cleaning and stratification of training data”), provides high-precision building outlines derived from Sentinel-1 and Sentinel-2 imagery in 2019. To estimate building height, we incorporated two global datasets: GHS-BUILT-H R2022A and 3D-GloBFP (Che et al., 2024). The 3D-GloBFP integrates remote sensing data and building morphology using an extreme gradient boosting (XGBoost) model. It is a vector dataset that provides building footprints with height attributes. We then rasterized 3D-GloBFP to 100 m resolution for alignment with GHS-BUILT-H R2022A. Although both datasets offer global coverage, we observed gaps in artificial object representation within 3D-GloBFP. To mitigate this issue, we gap-filled missing buillding in rasterized 3D-GloBFP layers using corresponding GHS-BUILT-H R2022A values.

Ancillary remote sensing and topographic data: In addition to DSMs and structural data, we incorporated optical remote sensing indices and topographic derivatives to refine terrain modeling. Long-term Normalized Difference Vegetation Index (NDVI), Normalized Difference Wetness Index (NDWI), and Near-Infrared (NIR) reflectance were included for the periods 2006–2010 and 2011–2015. These temporal windows align with the acquisition periods of AW3D30 and GLO30, ensuring consistency in object removal. The indices were derived from Landsat Analysis Ready Data (GLAD ARD) (Potapov et al., 2020), a harmonized dataset integrating imagery from Landsat 5 TM, Landsat 7 ETM+, and Landsat 8 OLI/TIRS. For each time period, we removed cloud from the scenes, aggregated the available images and computed the 5th, 50th, and 95th percentiles for each index to capture the distribution of surface conditions over time. Additionally, we incorporated topographic information from ETOPO2022 (MacFerrin et al., 2024), an updated global topographic-bathymetric dataset at a 15-arc-s resolution. ETOPO2022 integrates various data sources, including bare-earth DEMs, sea and lake bathymetry, and ice-sheet bed elevation. Its land topography component is derived from aggregated FABDEM and GLO30, making it suitable for global DTM applications. From this data set, we extracted slope information to provide a gentle steepness.

Table 1 A list of covariates for global terrain modeling.

Dataset	Variables	Grid spacing	Temporal range	Reference	
Copernicus GLO-30 Digital Elevation Model	Elevation height	1 arc s	2011–2015	European Space Agency (ESA) (2024)	
ALOS World 3D-30 m (AW3D30) dataset	Elevation height	1 arc s	2006–2010	Tadono et al. (2014), Takaku, Tadono & Tsutsui (2014)	
Global 2010 Tree Cover	Tree cover	1 arc s	2010	Potapov et al. (2022)	
Global Forest Canopy Height, 2019	Canopy height, Canopy edge	1 arc s	2019 (April–October)	Potapov et al. (2021)	
ETH Global Sentinel-2 10 m Canopy Height (2020)	Canopy height, Canopy edge	1/3 arc s	2020	Lang et al. (2023)	
Landsat Analysis Ready Data (GLAD ARD)	NDVI, NDWI, NIR (p05, p50, p95)	1 arc s	2006–2010, 2011–2015	Potapov et al. (2020)	
Earth Topography 2022 (ETOPO 2022)	Slope in degree	15 arc s	Data collected from various source	MacFerrin et al. (2024)	
World Settlement Footprint 2019 (WSF2019)	Building outline	1/3 arc s	2019	Marconcini et al. (2021)	
GHS-BUILT-H R2022A	Building height	3 arc s	2018	Schiavina et al. (2022)	
3D-GloBFP	Building height	Vector polygon	2020	Che et al. (2024)	

Production of global ensemble DTM

Global-to-local transfer learning modeling for DTM prediction

A DEM is a general term that encompasses both DSMs and DTMs. A DSM represents the Earth’s surface, including all above-ground objects such as buildings and trees. In contrast, a DTM represents “bare earth”, without natural and human-made objects (Guth et al., 2021). The primary distinction between DSMs and DTMs lies in their application: DSMs are often used for telecommunication or other line-of-sight planning, while DTMs are used for simulating natural processes, such as hydrological modeling, landform classification, and geological structure analysis. In this study, we focus on modeling the Earth’s terrain, employing the best available methods to remove tree canopy and other surface objects from the DSM. In other words, we implemented a direct modeling of the terrain from DSMs to a DTM instead of e.g., fitting a model to estimate canopy height, then derive canopy height from a DSM. This approach is different from previous global DTM mapping research, which focuses on object detection and removal to minimize the alteration of the bare earth surface (Dusseau, Zobel & Schwalm, 2023; Hawker et al., 2022; Pronk et al., 2024; Uhe et al., 2025; Yamazaki et al., 2017). In contrast, our direct modeling approach adopts Random Forest (RF) to merge multiple DSMs as base input maps, and purposively add layers that represent or reflect the aboveground objects and terrain complexity. In addition, we adopt a framework to train a global-to-local model in order to address the global diversity of land cover and landform. Global-to-local modeling is a two-step transfer learning training framework: the global model trains with representative, globally stratified sampling, and the local model inherits the global model, training with additional locally stratified sampling. This framework not only preserves the generalization of the global model, but also reinforces the local features to achieve better performance (Fig. 2).

Figure 2 Visual explanation of the global-to-local transfer learning modeling.

The system consists of M models where M corresponds to the number of tiles + 1 global model.

We first trained a global RF model using 10% random global samples for hyperparameter tuning, reducing the processing time and keeping the model representative of the whole data set. The remaining 90% samples were used in cross-validation. After model fine-tuning, all samples were then used to train a global RF model with 100 decision trees. This model was used as a starting point for training locally enhanced models with additional global LiDAR terrain height samples (transfer learning), selected according to 5° × 5° tiles. For each tile, additional samples, derived according to our global modeling stratification strategy (see “Cleaning and stratification of training data”), were used for hyperparameter tuning and training of 100 additional decision trees. Locally enhanced models, with 200 decision trees in total, were trained for all tiles that met the minimal training sample requirement ( ∼average 10 points in each vegetation stratum); otherwise, the global model was used to run DTM predictions. Our predictions estimated the mean and standard deviation of DTM elevation considering all individual decision trees of the final RF models, implemented with the library scikit-learn (an extend version of the class RandomForestRegressor).

Lastly, the prediction outputs were filtered by a adaptive filter (Gallant, 2011) and bilateral filter (Tomasi & Manduchi, 1998) to remove speckle noise from the data fusion. The filters are also used in FABDEM and MERIT DEM (Hawker et al., 2022; Yamazaki et al., 2017). We used the functions adaptive_filter, with 5 × 5 neighborhood size, and the function bilateral_filter, setting the standard deviation distance to 2 and the standard deviation intensity to 3. The functions are implemented in Whitebox Workflows and the filters were only applied on the tree and building cover pixels (Global 2010 Tree Cover is not 0 and WSF2019 is not 0), in order to avoid potential artifacts in non-aboveground-object areas.

Model evaluation

We validated our models and predictions using independent reference datasets and using visual inspection and evaluation metrics. The quality of the DTM is not only the accuracy of pixel values to the reference, but also the consistency of the terrain surface. Therefore, the validation focuses on not only the accuracy metrics, but also the cross section, spatial visualization on DTM and hillshading. We incorporated independent source of validation datasets including: High quality LiDAR datasets: A collection of reference LiDAR DTMs (Guth et al., 2023) used by Bielski et al. (2024), Guth et al. (2024) for the evaluation of global DEMs. The reference DTM locations are mainly in Europe, North and South America, and completely under-represent Asia and Africa.

Global distributed Global Navigation Satellite System (GNSS) stations: Altitude data from GNSS stations collected from Nevada Geodetic Laboratory (geodesy.unr.edu/velocities/midas.IGS14.txt). The data set is global coverage and is also used to validate the Global Digital Elevation Merged Model 2024 (GDEMM2024) (Ince, Abrykosov & Förste, 2024).

In addition to the validation dataset, three metrics are employed to measure the prediction performance: the root mean square error (RMSE), the mean absolute error (MAE), and their mean error (ME):

(1) RMSE=1NΣi=1N(y^-y)2,

(2) MAE=1NΣi=1N|y^-y|,

(3) ME=1NΣi=1N(y^-y).

RMSE is more sensitive to outliers, whereas MAE measures individual differences equally weighted; ME measures the prediction bias, indicating whether the model overestimates or underestimates the height of the terrain. In addition to point-to-point metrics, for comparing with raster reference DTMs about preserving DTM structure, we employed structural similarity index measurement (SSIM) (Wang & Bovik, 2009; Wang et al., 2004).

Multiscale terrain variables derivation

Pyramid representation of DTM

Multiscale terrain variables, also called terrain attributes, were derived from the GEDTM30, which is crucial to modeling soil, water, and biological processes (Behrens et al., 2018; Kulp & Strauss, 2018; Lozbenev et al., 2021; Moore, Grayson & Ladson, 1991). The multiscale process comprises two steps: (1) the DTM is down-scaled to coarse resolution, and (2) the original layer and the down-scaled DTM layers run the terrain variables derivation independently (Fig. 3). In our research, the objective is to produce the terrain variables in grid spacing of 30, 60, 120, 240, 480 and 960 m, increasing scale by a factor of 2. We ran the derivation using Whitebox Workflows version 1.3.4 (WbW) in Python (https://www.whiteboxgeo.com/whitebox-workflows-for-python/). We set identical arguments for each function on different resolutions to make all derivation consistent. For resampling from coarser to finer resolution, we recommend using cubic splines interpolation. Note that we have initially tested derivation in SAGA GIS, GRASS GIS and GDAL (all provide extensive libraries for terrain variables derivation), however, Whitebox Workflows have shown to be fastest and easiest to implement on large grids.

Figure 3 Pyramid representation of terrain variables derivation.

Above: GEDTM30 at three scales; below: corresponding slope maps generated using WhiteboxTools.

Terrain variables

There are two types of terrain variables: regional and local (Olaya, 2009b; Shary, 1995; Wilson, 2012; Wilson & Burrough, 1999). The total number of information that can be derived from a relief model is extensive and probably counts in hundreds of variables. We selected 15 fundamental terrain variables that represent measures of local topographic position, curvature, hydrology, incoming solar light and shadow. Local terrain variables include hillshade, slope, openness, curvature and topographic position such as deviation from mean elevation and spherical standard deviation of normals. They are introduced below: Hillshade: Hillshade, also called shaded relief, is a terrain visualization technique that accounts for the positioning of a theoretical light source and the local slope and aspect of the elevation surface (Horn, 1981). In this research, we derived mutlidirectional hillshade, which integrates hillshade values calculated from multiple sources of illumination. We used the function MultidirectionalHillshade and set altitude to 30 degrees, and Z-factor to 1 as default.

Slope in degree: The slope gradient, measured in degrees, measures the steepness of the terrain surface in each grid cell in a DEM. The algorithm used to calculate the slope used the third-order bivariate Taylor polynomial method described by Florinsky (2016). We used the function slope, which applies a polynomial fit of the elevations within the 5 × 5 neighborhood surrounding each cell, to mitigate against the outlier elevations.

Openness: The dominance or enclosure of a location on an irregular surface. It has two viewer perspectives, positive and negative openness. The mean value of all zenith angles gives positive openness, while the mean nadir value gives negative openness (Yokoyama, Shirasawa & Pike, 2002). In our research, we used the function openness applied the formula from Yokoyama, Shirasawa & Pike (2002) to derive both positive and negative openness. We set search distance to three grid cells.

Spherical standard deviation of normals: The measure of the angular dispersion of the surface within a local neighborhood of a specified size. It indicates the complexity, texture, and roughness of the surface shape. Grohmann, Smith & Riccomini (2010) discovers that the vector dispersion, a related measure of the angular dispersion, increases monotonically with scale. The scale relation can therefore be estimated by isolating the amount of surface complexity associated with specific scale ranges: large scale reflects the texture; small scale reflects the complexity. We used the function spherical_std_dev_of_normals and the local neighborhood size was set to 3 in our research.

Difference from mean elevation: The calculation of the difference between the elevation of each DEM pixel and the mean elevation within a moving window of a size. It is also a measure of local topographic position. In our research, we used the function difference_from_mean_elevation and set both filterx and filtery to 3.

Maximal curvature: The curvature of a principal section with the highest value of curvature at each DEM pixel. The positive values correspond to ridge positions while negative values are indicative of closed depression (Florinsky, 2016). It is measured in unit m−1. We used the function maximal_curvature, applied the formula from Shary, Sharaya & Mitusov (2002), with a polynomial fit of elevations within the 5 × 5 neighborhood surrounding each cell. Log-transform is adopted.

Minimal curvature: The curvature of a principal section with the lowest value of curvature at each DEM pixel. The positive values correspond to hills while negative values are indicative of valley positions. It is measured in unit m−1. We used the function minimal_curvature, applied the formula from Shary (1995), Shary, Sharaya & Mitusov (2002), with a polynomial fit of the elevations within the 5 × 5 neighborhood surrounding each cell. Log-transform is adopted.

Profile curvature: Profile, or vertical curvature (unit: m−1) is the curvature of the rate of change in slope along a flow line. Positive values correspond to flow acceleration, while negative values are indicative of flow deceleration. It is the curvature of a normal section having a common tangent line with a slope line on each DEM surface. We used the function profile_curvature, applied the formula from Shary (1995), Shary, Sharaya & Mitusov (2002), with a polynomial fit of the elevations within the 5 × 5 neighborhood surrounding each cell. Log-transform is adopted.

Tangential curvature: Tangential, or horizontal curvature (unit: m−1) is the curvature of an inclined plane perpendicular to both the direction of flow and the surface. The positive values correspond to flow divergence while negative values are indicative of flow convergence (Wilson & Gallant, 2000). We used the function tangential_curvature, applied the formula from Shary (1995), Shary, Sharaya & Mitusov (2002), with a polynomial fit of the elevations within the 5 × 5 neighborhood surrounding each cell. Log-transform is adopted.

Ring curvature: The product of horizontal excess and vertical excess. Ring curvature (unit: m−2) has values equal to or greater than zero, and high values indicates the greater potential of the flow motion. It is used to measure flow line twisting (Shary, 1995). We used the function ring_curvature, applied the formula from Shary (1995), Shary, Sharaya & Mitusov (2002), with a polynomial fit of the elevations within the 5 × 5 neighborhood surrounding each cell. Log-transform is adopted.

Shape index: The shape indicator. It is independent of the size i.e., the amount of curvature, as distinct from the type of curvature. The shape index represents the intuitive notion of local shape, with positive values indicative of convex landforms, negative values corresponding to concave landforms (Koenderink & Van Doorn, 1992). We used the function shape_index applied the formula from Koenderink & Van Doorn (1992), with a polynomial fit of the elevations within the 5 × 5 neighborhood surrounding each cell.

The regional terrain variables consider other parts of the DEM or the entire scene in addition to the exact point where they should be calculated (Olaya, 2009a). We selected mainly hydrological variables that require flow enforcement to ensure that interior topographic depressions and flat areas have been removed from the input DEM. The depression-breaching algorithm used for flow enforcement in this work was implemented based on Lindsay (2020). The approach uses a least-cost path analysis to identify the breach channel that connects pit cells (i.e., grid cells for which there is no lower neighbor) to some distant lower cell. We used the function BreachDepressionsLeastCost, which combines breaching at first and filling at the second. We set the maximal search distance to 100 pixels, and used an increment value of 0.001 to connect input DTM pixels within depressions to lower cells. Any depressions that could not be resolved using a 100-cell-long breach channel were subsequently resolved using depression filling. After preprocessing the DTM, we produced specific catchment area, slope length and steepness factor, and topographic wetness index: Specific catchment area: The upstream catchment area of a unit contour length. In our research, we computed SCA using Qin et al.’s (2007) method. It relates the degree of flow dispersion from a grid cell to the local maximum downslope gradient. We used the function QinFlowAccumulation and set upper-bound exponent and slope to 10 and 45 respectively.

Slope length and steepness factor: One of the components used to estimate the physical potential for sheet and rill erosion in upland catchments (Moore, Grayson & Ladson, 1991), so-called length-slope factor (LS factor) or sediment transport index. The inputs of LS factor are slope in degree and SCA which are computed in advance and described above. We used the function sediment_transport_index and set SCA exponent to 0.4 and slope exponent to 1.3. Log-transform is adopted for the result.

Topographic wetness index: The index describes the propensity for a site to be saturated to the surface given its contributing area and local slope characteristics. It is commonly used in the TOPMODEL rainfall-runoff framework (Beven, 1997). The inputs of TWI are slope in degree and SCA which are computed in advance. We use the function WetnessIndex. Log-transform is adopted.

The methods that were adopted in the derivation functions and their details are documented in, and all right are reserved ©WhiteboxTools user manual (Lindsay, 2018).

Optimizing of terrain variables derivation

To speed up derivation of terrain variables and cut-down costs of derivation, the input DTM is split into tiles then run in parallel. To preserve the correctness of the terrain variables, the input DTM should has same vertical and horizontal units, where Equi7 has been commonly used to cope with global-scale derivation (Amatulli et al., 2020). In addition, the redundant pixels of DEM should be masked out to improve the efficiency of derivation. Overall, to optimize the derivation in high resolution of the world, data parallelization and removal of redundant input DTM pixels are key to reducing the computing costs. We next describe how we selected a consistent land mask and how exactly we implemented optimized parallelization.

Preparation of land mask

We created an ensemble land mask based on GLO30, AW3D30 and ESA World Cover. The three products so far have the most complete land coverage, and they are complimenting each other in creating the most inclusive land mask. The World Cover of ESA is a global land cover product for 2020 and 2021 at 10 m resolution (Zanaga et al., 2022) and contains the finest land detail, but inland water is defined as the same class as ocean. Masking the water would end up losing the inland water pixels. GLO30 and AW3D30 are DSMs (described in “Covariates”). Both cover the whole period of 2006–2015, being possible to capture the emergent islands or submerged coastline due to sea level rise in this period. However, these DSMs define the ocean as sea level (altitude =0) which could also happen at inland area and lead to losing inland area pixels where its elevation is equal to 0. Finding the union of “not water” for the ESA World Cover and “not at sea level” for GLO30 and AW3D30 is the solution to create the only ocean mask land mask.

To produce a consistent land mask we aggregated ESA world cover to 1 arc s using mode as resampling method in GDAL to match the grid spacing of GLO30 and AW3D30. If the pixel reaches the condition where ESA World Cover is not water, or GLO30 is not 0, or AW3D30 is not 0, the pixel would be identified as land, and set to value 100. The other pixels are then set to 0 and saved as no data. In addition, the definition of the coastline could be controversial to countries, and a pixel can also not be recovered once it has been masked. Therefore, we decided to apply a buffer to the only-ocean-mask land mask. The buffer was created by aggregating the 1-arc-s mask to 30-arc-s grid spacing by maximum, to extend the land ∼1 km at the coastline. Subsequently, the 30-arc-s mask was then down-scaled back to 1-arc-s grid spacing by cubic spline. Ultimately, we filtered all the pixels with value (not 0) to 100 and defined 100 as the land pixel, and defined 0 as the ocean pixel. The land mask was used in the production of GEDTM30 and also in the production of 15 derived terrain variables.

Tiling system and projection

We implemented data parallelization followed by the defined Equi7 tiling system (Bauer-Marschallinger, Sabel & Wagner, 2014). Equi7 is a continent-based projection system that minimizes data oversampling over global land surface. However, if we would implement the original Equi7 tiling system to derive variables without any overlap, final mosaics would eventually show tile-boundary effects. To resolve this issue, a spatial overlap is required to create seamless global maps which are artifact-free and hydrologically connected. The previous article published by Amatulli et al. (2022) proposed an irregular tiling system, which retains the entire drainage basin in each tile to preserve the lateral and longitudinal connectivity. Such an irregular tilling system contains instances like partially included drainage basins across multiple tiles, or the tile could not be enlarged beyond the maximum number of grid cells. The irregular tiling system used by Amatulli et al. (2022) is in our opinion not generalizable to multiscale terrain variables derivation. Instead, we implemented an extended tiling system with consistent tiles optimized for hydrological modeling. We considered three aspects of parallelization: Size of tiles;

Overlap of tiles;

Software implementation (aiming at optimized computing and use of RAM) by using S3 (Amazon’s Simple Storage Service) and InfiniBand to access large number of tiles in parallel;

After initial testing, we decided that the 600 km by 600 km Equi7 tiles could be ideal as the tiling basis for derivation. To derive the local terrain variables we padded 1.92 km on each Equi7 tile, which preserves at least two pixels from all sides in the coarsest resolution (960 m) input DTM; Regional terrain variables instead need to consider the whole scene of the input data, so larger overlap is required. We discovered that padding with 16%, which is about 102 km in 1 arc s input DTM, is sufficient to mitigate the boundary effect. Although the extended tile caused around 1.78 times of data operation toward size and time, it ensures the hydrological connectivity persist until the boundary of the Equi7 tile.

In practice, the input GEDTM30 was first reprojected onto the Equi7 projected coordinate system (geographic to projected coordinate system) by cubic spline resampling approach, and cropped in extended tiles for the regional terrain variables. Subsequently, each tile was aggregated to multiscale resolution in 60, 120, 240, 480 and 960 m by average using GDAL. The original 30 m tile applied a Gaussian filter to reduce noise. Aftermath, the DTM tiles ran through the regional derivation function sequentially, with the breaching and depression-filling procedure applied before the hydrological analysis. Aftermath, the processed local and regional variables were truncated back to 600 km by 600 km with 960 m padding. Table 2 summarizes the overall information of the 15 terrain variables mentioned in “Terrain Variables” with tile overlapping distance.

Table 2 Product list of terrain variables.

Product name	Type	Category	Tile overlap	
Slope in degree	Local	Landform	1.92 km	
Hillshade	Light and Shadow	1.92 km	
Positive openness	1.92 km	
Negative openness	1.92 km	
Minimal curvature	Curvature	1.92 km	
Maximal curvature	1.92 km	
Profile curvature	1.92 km	
Tangential curvature	1.92 km	
Ring curvature	1.92 km	
Shape index	1.92 km	
Deviation from mean elevation	Topographic position	1.92 km	
Spherical standard deviation	1.92 km	
Specific catchment area	Regional	Hydrology	101.76 km	
Slope length and steepness factor	101.76 km	
Topographic wetness index	101.76 km	

The conversion between the Equi7 and WGS84 coordinate systems requires special consideration near the 180th meridian. The Equi7 grid is continent-based, adhering to the boundaries of continental landmasses, whereas the WGS84 system uses a global reference ellipsoid with a defined longitudinal boundary at the 180th meridian. During reprojection, if an Equi7 tile spans the 180th meridian, the reprojected tile in WGS84, similar to a polygon, would contain the few vertices approximate to 180∘ and also some approximates to −180∘ on the other side. Creating a bounding box from these reprojected tiles for cropping input GEDTM30, the longitudinal range would extend across the entire world from −180∘ to 180∘. Consequently, the attempt to read the huge bounding box of the 1-arc-s input DTM results in dramatically slowing down the process or even excessive memory usage. To address this issue, a dedicated set of irregular tiles in W84 was generated and reprojected into the Equi7 system. These irregular tiles are strategically located to mitigate boundary effects in variables and ensure comprehensive data coverage. The issue is particularly pronounced in regions such as the Chukotka Autonomous Okrug in Russia, the North Island of New Zealand, and several Pacific islands. Additionally, the Equi7 Oceania tile system fails to adequately cover specific Pacific territories, including Nouvelle-Calédonie, Yap (Micronesia), Palmerston Island, etc. To resolve this, additional irregular tiles are created in the EPSG:4326 coordinate system, ensuring seamless projection transformations without exceeding the 180th meridian boundary and providing full coverage of these regions. Figure 4 illustrates the tiling system, including regular and irregular tiles that resolve the issues of the 180th meridian and missing islands.

Figure 4 Equi7 tiling system for six continents and irregular tiles for optimizing terrain variables derivation of the world as shown in the WGS84 geographic coordinate system.

Base map © OpenStreetMap.

Results

Global ensemble digital terrain model (GEDTM30)

Global DTM production result

Figure 5 shows the distribution of the tiles whether local models were produced. As a result, the global-to-local model is used to predict 1,157 land tiles in 5∘ ×5∘ grid. Ultimately, 1,019 local models have been fitted, where 138 tiles do not have sufficient samples for the local models. The red tiles were only predicted by global models due to insufficient amount of training data. The background is the land mask that was specially created to buffer the coastline. It also shows the promising result that covers small Pacific islands such as Nauru and French Polynesia and covers until the northernmost point of land.

Figure 5 Distribution of GEDTM30’s global and local tiles on the ocean-only land mask with a zoom-in on two tiles with smaller islands.

Assessment of global-to-local modeling

The analysis conducted for all reference DTMs demonstrates a reduction in the average RMSE from 4.14 to 3.86 m, representing an approximate 9% improvement when comparing the DTM generated using only the global model to that produced by the global-to-local model. Figure 6 illustrates an visual example of the differences between the predictions and the reference DTM. Assessment of the global-to-local model demonstrates an improvement in local accuracy while maintaining a consistent terrain representation. To evaluate the impact of incorporating additional local samples, we generated predictions using both the global model and the global-to-local (locally enhanced) model for an area in Trentino, Italy. Two tiles were produced and compared against the reference DTM. In summary, this shows that the locally enhanced model exhibits improved accuracy. The upper histogram in Fig. 6 shows that the locally enhanced model reduces both error and bias compared to the global model alone. The lower map highlights how additional local samples contribute to improving accuracy while maintaining the consistency of the terrain.

Figure 6 Visual comparison of global vs. global-to-local models.

This shows that additional local training data improves prediction producing a lower difference between the reference model and GEDTM30. Location shown: Trentino, Italy.

Accuracy comparison with source DSMs

We further evaluated GEDTM30 against source DSMs (GLO30 and AW3D30) to assess its ability to remove vegetation and artificial objects while improving accuracy. First, we used a multidirectional hillshade (WhiteboxTools QGIS Plugin, 45-degree altitude) to visually inspect the efficiency of object removal. Figure 7A illustrates the removal of forest and vegetation objects in Departamento de Artigas, Uruguay. The red circle and arrows highlight a tree patch visible in the Google Satellite image, which is effectively removed in GEDTM30. Furthermore, GEDTM30 integrates the roughness characteristics of GLO30 and AW3D30. Figure 7B demonstrates the removal of artificial objects in São Paulo, Brazil, where buildings are effectively removed while topographic characteristics are preserved.

Figure 7 Visualization of object removal from DSMs to DTM.

(A) Forest/vegetation object removal of Departamento de fig: Artigas, Uruguay; (B) building/artificial object removal of São Paulo, Brazil. Map data © Google Satellite.

Next, Fig. 8 presents a comparative evaluation of object removal against reference DTMs. We evaluated GEDTM30, GLO30 and AW3D30 using cross-sectional analysis, visual inspection, and error distribution analysis. The left panel (cross section) of Fig. 8 displays terrain roughness consistency and noise levels between DEMs. The middle panel provides a full-scene visualization of object removal and modeled terrain consistency, in order to inspect occasional model-induced hallucinations arose due to optimization in machine learning methods (Guth et al., 2024). The right panel presents error distributions via histograms, highlighting the effectiveness of object removal. For removal of vegetation objects, we extracted the pixel that falls on the canopy cover 10–50% and the canopy cover >50% for comparison. For the removal of artificial objects, we mask the pixels where WSF2019 classified areas as non-artificial objects. Figure 8A focuses on forest/vegetation object removal in Departamento de Artigas, Uruguay and Fig. 8B focuses on building/artificial object removal in São Paulo, Brazil.

Figure 8 Comparison of DEMs in cross section, visual inspection and distribution of error.

(A) Forest/vegetation object removal against reference DTM in Departamento de Artigas, Uruguay; (B) building/artificial object removal against reference DTM in São Paulo, Brazil. Map data © Google Satellite.

Third, GEDTM30 is validated with reference DTMs using evaluation metrics, RMSE, MAE, and ME. We compared with the source DSMs, GLO30 and AW3D30 at the visual inspected sites (Fig. 8), São Paulo for artificial removal and Departamento de Artiga, Urugauy for vegetation removal. Table 3 summarizes the accuracy metrics grouped by inspection sites and land use and cover (LULC). The results show that GEDTM30 has the best RMSE, MAE and ME of all groups, and has approximately a 30% improvement in accuracy of GLO30 in the build-up and tree cover >50%.

Table 3 Accuracy evaluation of GEDTM30 with its source DSMs, GLO30 and AW3D30 for built-up and forest areas at the visual inspected sites of “Accuracy comparison with source DSMs”.

Dataset	Value pixels	Land cover	RMSE	MAE	ME	
GEDTM30	13,424	Tree cover >50%	3.73	1.91	−0.00	
GLO30			5.14	3.78	−2.94	
AW3D30			6.17	5.05	−4.53	
GEDTM30	12,607	Tree cover 10–50%	2.87	1.56	0.09	
GLO30			3.19	1.85	−0.73	
AW3D30			4.25	2.83	−2.34	
GEDTM30	341,292	Built-up	3.03	2.21	−1.92	
GLO30			4.06	3.19	−3.06	
AW3D30			7.45	5.64	−5.62	
Note:

The lowest RMSE, MAE, and ME among all groups as a DTM are shown in bold.

Accuracy comparison with other global DTMs

GEDTM30 was also compared with other global DTMs, including MERIT DEM, FABDEM, and FathomDEM (Hawker et al., 2022; Uhe et al., 2025; Yamazaki et al., 2017), across diverse landforms. We selected three representative study areas: Trentino, Italy (Fig. 9), Chincoteague, the United States, (Fig. 10), and Vestfold, Norway, (Fig. 11), representing mountainous region, flood plain, and boreal forest. The first column of these figures is the visualization of the reference DTM and the global DTMs. The second column is the difference (reference DTM-global DTM), and the third column is the mutlidirectional hillshade visualization.

Figure 9 Existing DTM products comparison in mountainous region (Trentino, Italy).

Figure 10 Visual comparison of existing DTM products for a selected flood plain (Chincoteague, the United States).

Figure 11 Visual comparison of existing DTM products comparison in boreal forest (Vestfold, Norway).

In Trentino (Fig. 9), all DTMs have performed better in the valley surrounded by mountains, which is relatively flat. As the slope increases, MERIT DEM has a significantly higher terrain estimate than other two models, and FABDEM also overestimate the terrain height/underestimate the tree height significantly in the steep area in the scene. GEDTM30 and FathomDEM have a rather balanced result. However, GEDTM30 slightly overestimates the steep area and underestimates the high attitude area in the bottom right corner and top left corner of the scene. On the right side of Fig. 9, mutlidirectional hillshade shows GEDTM30 is slightly noisier/softer than FABDEM and FathomDEM as the same 1-arc-s grid spacing DTMs. Finally, FABDEM appears to show pit holes on the slope, locating at the left and upper side of the valley, which could be due to the issues in the height removal algorithm used.

In Chincoteague (Fig. 10), a flat coastal plain, we incorporated land-use/land cover map change from GLAD for the period 2000–2021 (Potapov et al., 2022) to collate the difference map. The primary land use and land cover here is the build-up area (cyan), the stable crop field (orange), and the stable tree cover (green). FABDEM and GEDTM30 have a more significant error in stable tree cover than in built-up area and stable crop field. GEDTM30 does not remove some tree patches, leading to an overestimate of the terrain height, whereas FABDEM has a more extreme error than GEDTM30. It appears that FathomDEM performs the best in object removal and has the smallest error. Ultimately, visual inspection of all DTMs using mutlidirectional hillshade, shows that FathomDEM appears to be the most accurate containing the least artifacts, followed by MERIT DEM, and then GEDTM30 and FABDEM.

Vestfold, Norway (Fig. 9) is a boreal forest region with rugged terrain. The difference against the reference DTM is generally smaller than in Trentino (Fig. 9), but both GEDTM30 and FABDEM tend to underestimate terrain height, whereas MERIT overestimates elevations. FathomDEM hence has a rather balanced result. Overall, the high error region, same as Trentino, locates at steeper area. Fathom remains the smallest error among the target DTMs. The terrain roughness in GEDTM remains the greatest however.

Table 4 summarizes the accuracy metrics of the above three sites. The results show regardless of the locations, show the consensus of better accuracy in FathomDEM, followed by GEDTM30 and then FABDEM and MERIT DEM. In addition, to better estimate the model performance on different land cover, we aggregated the pixels from all the reference DTM sites and grouped them by land use and land cover. Table 5 summarizes the result of the evaluation metrics on various stable LULC classes aggregated from GLAD Global LULC Change (Potapov et al., 2020). As a result, FathomDEM performs overall the best in cropland, built-up, forest areas, whereas FABDEM seems to have a lower error in bare earth and sparse grassland. GEDTM30, FathomDEM, and FABDEM all maintain mean errors (ME) under 1 m for most of the LULCs, except for dense short vegetation in GEDTM30. On average, Fathom has ∼25% smaller RMSE than GEDTM30.

Table 4 Accuracy evaluation of target DTM at the visual inspected sites of “Accuracy comparison with source DSMs”.

Location	Dataset	Valid pixels	RMSE	MAE	ME	SSIM	
MERIT DEM	Trentino, Italy	182,618	13.20	9.98	−5.45	0.9927	
FABDEM			8.93	6.29	−3.32	0.9970	
FathomDEM			5.45	3.82	−1.29	0.9983	
GEDTM30			7.09	5.08	−1.83	0.9975	
MERIT DEM	Chincoteague, the USA	880,611	2.08	1.63	−1.35	0.7005	
FABDEM			3.33	1.77	−1.43	0.6887	
FathomDEM			0.85	0.57	−0.24	0.8571	
GEDTM30			2.14	1.40	−0.72	0.6679	
MERIT DEM	Vestfold, Norway	2,563,774	5.21	3.83	−1.49	0.9786	
FABDEM			4.45	3.11	0.82	0.9842	
FathomDEM			2.28	1.59	0.41	0.9948	
GEDTM30			3.47	2.64	1.68	0.9919	
Note:

Results with the best accuracy are highlighted in bold.

Table 5 The evaluation metrics of target DTMs on various stable LULC classes from aggregating all reference LiDAR DTM sites.

Dataset	Land cover (c)	Land cover (%)	RMSE	MAE	ME	
MERIT DEM	Cropland, stable	11.3	2.31	1.79	−0.91	
FABDEM			1.65	1.03	0.34	
FathomDEM			1.45	0.82	0.39	
GEDTM30			1.56	0.95	0.38	
MERIT DEM	Built-up, stable built-up	12.5	4.76	3.19	−1.79	
FABDEM			2.74	1.67	0.32	
FathomDEM			1.95	1.23	0.50	
GEDTM30			2.75	1.75	0.04	
MERIT DEM	Terra Firma, true desert	5.3	6.16	4.44	−2.82	
FABDEM			1.81	0.71	0.02	
FathomDEM			1.89	0.93	0.17	
GEDTM30			2.65	1.56	−0.09	
MERIT DEM	Terra Firma, stable tree cover	15.0	10.76	7.56	−2.12	
FABDEM			6.82	4.30	−0.25	
FathomDEM			4.95	2.87	−0.18	
GEDTM30			6.72	4.38	0.98	
MERIT DEM	Terra Firma, semi-arid	20.3	6.63	4.73	−1.52	
FABDEM			2.37	1.00	0.24	
FathomDEM			2.52	1.23	0.27	
GEDTM30			3.38	1.97	0.67	
MERIT DEM	Terra Firma, dense short vegetation	15.9	5.56	3.53	−0.51	
FABDEM			2.65	1.18	0.34	
FathomDEM			2.50	1.20	0.64	
GEDTM30			3.35	1.90	1.13	
Note:

Results with the best accuracy are highlighted in bold.

Beyond remote sensing-derived DTMs, we further validated models using independent, globally distributed GNSS station data. Figure 12 illustrates the distribution of the GNSS stations and the comparison of heights of GNSS stations and target DTMs. Figure 12A is the distribution of the GNSS stations and the elevation of each record in orthometric height. Figure 12B presents a histogram of GNSS-DTM elevation differences (GNSS height-DTM). The legend indicates accuracy assessments among the global DTMs. GEDTM30 achieves the best mean at 7.34 m, standard deviation at 7.77 m, and RMSE at 10.69 m when compared to GNSS stations. However, systematic discrepancies persist between remote sensing DTMs and GNSS measurements, as previously reported by Ince, Abrykosov & Förste (2024) in their validation.

Figure 12 Accuracy comparison using the GNSS stations height around the world.

(A) The distribution of GNSS station extracted from NGL; (B) the histogram of height difference and metrics (mean, standard deviation, RMSE) among global DTMs. Base map © Google Satellite.

Uncertainty map of terrain height prediction

Figure 13 presents the global uncertainty map for predicting terrain height. The results indicate that uncertainty is closely linked to steepness and canopy density of terrain. The upper plot in Fig. 13 illustrates the global standard deviation of the terrain height predictions, revealing areas of high uncertainty primarily in dense forest regions such as the Amazon and the Congo basin, as well as major mountain ranges such as the Himalayas, Andes, and Alps.

Figure 13 The uncertainty of terrain of the world with subplots: (A), slope and uncertainty map in Camargo, Bolivia; (B), tree cover and uncertainty in Linte, Cameroon.

Map data © Google Satellite.

Figure 13A provides a detailed view of the Andes mountain range in Camargo, Bolivia, where terrain slope derived from GEDTM30 (bottom-left corner) exhibits a pattern similar to the standard deviation of the height predictions. Figure 13B in the meanwhile zooms in on the Northern Congolian forest-savanna mosaic ecoregion in Linte, Cameroon, demonstrating a correlation between terrain prediction uncertainty and the GLAD tree cover map (Potapov et al., 2020).

Terrain variables derivation result

Software testing and optimization

This section summarizes the results of optimization test and the estimated computation time for global terrain variables derivation. The test was carried out by running the 60 m grid spacing derivation for all tiles (operation details in “Optimizing of terrain variables derivation”). The processing time for each tile and process is recorded. Figure 14 illustrates the average individual and cumulative processing time of terrain variables derivation in 60 m grid spacing. According to the Fig. 14, breaching depression takes the longest average processing time at 194 s per tile and the topographic wetness index takes the shortest to complete at the average processing time of 5 s. The complete computational workflow averages 742 s per 600 km by 600 km Equi7 tile at 60 m grid spacing. Extrapolating from this, a 30 m resolution grid spacing is estimated to complete within two days, using 1,344 CPUs on 14 servers equipped with 1 TB of RAM each and connected to S3 storage servers with InfiniBand. Using this infrastructure, the possible update for the multi-resolution variables can be completed in less than 60 h.

Figure 14 Individual and cumulative processing times for terrain variables derivation at 60 m grid spacing, averaged across all tiles.

Gallery of derived terrain variables

A total of 15 GEDTM30-derived terrain variables were generated, covering aspects of topographic position, light and shadow, landform characteristics, and hydrology. Figure 15 shows an example of all variables at a grid spacing of 1 arc s in Virunga National Park, located at the intersection of Rwanda, Uganda, and the Democratic Republic of the Congo (DRC). The variables include positive openness, negative openness, slope in degree, profile curvautre, tangential curvature, mutlidirectional hillshade, minimal curvature, maximal curvature, shape index, specific catchment aream topograhpic wetness index, ring curvauture, spherical standard deviation of normals (SphericalStdDevOfNormals), difference from mean elev (DiffFromMeanElev), and LS factor. These variables were generated using functions within Whitebox Workflows, with additional derivation details provided in the Materials and methods section.

Figure 15 A total of 15 terrain variables derived from GEDTM30 in Virunga National Park, Uganda, Rwanda, and DRC.

Map data © Google Hybrid.

Figure 16 additionally illustrates the pyramid representation of the topographic wetness index (TWI) at different grid spacing (30, 60, 120, 240, 480, and 960 m) in the Jhuoshuei River Basin, central Taiwan. The representation of the TWI pyramid in the alluvial plain highlights how the channel size and the distinctiveness of the cells evolve with increasing resolution. In the flat plain region shown in Fig. 16, the 30 m grid spacing TWI clearly differentiates between wet and dry pixels. As the size of the grid increases, the primary channel becomes more prominent and by the coarsest resolution (960 m), the flat plain exhibits a nearly uniform wetness distribution. This multiscale TWI representation effectively captures a gradient of wetness, where some cells remain consistently dry, others transition from dry to wet at coarser resolutions, and some are persistently wet across all scales.

Figure 16 Multiscale Topographic Wetness Index (TWI) at Jhuoshuei River Basin in centre Taiwan.

Base map © Google Earth Pro.

Consistency check

Figure 17 presents an assessment of spatial consistency across both processing tile and watershed boundaries, as well as a comparison with a leading global hydrological product, Hydrography90m (Amatulli et al., 2022). Two large drainage basins were selected for visual inspection: Lake Victoria in Africa (A) and Paraná Basin in South America (B). Watershed delineations are based on HydroBASINS Level 3 (Lehner & Grill, 2013), overlaid with the Equi7 processing tile grid in the upper panels of Fig 17. The lower panels display the TWI from GEDTM30 at the pixel level, alongside the corresponding TWI from Hydrography90m. The GEDTM30 TWI exhibits finer spatial detail due to its higher resolution, while maintaining a consistent spatial pattern with Hydrography90m. In addition, GEDTM30 preserves continuity not only across HydroBASINS watershed boundaries but also across processing tile boundaries.

Figure 17 Consistency check of GEDTM30 terrain variables using an example of TWI, comparing with Hydrography90m (Amatulli et al., 2022) at the processing tile and the watershed boundaries.

(A) Lake Victoria, Africa; (B) Paraná Basin, South America. Base map © Google Hybrid.

Discussion

Summary findings

We have developed a global, consistent and gap-free DTM, and an automated optimized procedure for deriving basic and advanced terrain variables based on a global 30 m DTM. Because we decided to use exclusively open data and open source licenses, our GEDTM30 can be considered a fully open data 1-arc-s grid spacing global DTM and geomorphometric library of morphometric and hydrological variables, with code available on Github and majority and inputs/outputs under CC-BY license from Zenodo or similar. Our outputs are general purpose but specially aim to serve various spatial predictive mapping projects e.g., described in Maxwell & Shobe (2022) and Behrens et al. (2018).

Our cross-validation accuracy assessment results show that the recently released FathomDEM (Uhe et al., 2025) still has a slightly smaller RMSE (on average 25% smaller) than our GEDTM30 in terms of vertical accuracy, although when compared using the GNSS points, the RMSE of our GEDTM30 is comparable to FathomDEM. A more comprehensive comparison of global DTMs, such as Guth et al. (2024), is still required to determine which global DTM has the best representation of the terrain, including features such as watershed boundaries, stream network, catchment area, and similar. Note that although we have looked at also integrating FABDEM and/or FathomDEM into the ensemble DTM, due to its NC-ND license and Fathom company requesting that their data be not used for producing open data (including DTM derivatives), we have finally decided not to mix FABDEM and/or FathomDEM with any of our outputs.

WorldDEM (5 m grid spacing; see some sample data in Fig. 18) is possibly even more interesting for users who require high-quality DTM data (Schrader et al., 2024), although we could only test some sample data since it is a commercial product and obtaining copies of the data for large areas would have been beyond our budget. We hope that our comparison of these data sets sheds a light on what is available to researchers and users and what the advantages are of each data set. We encourage users and data producers to register their terrain, LiDAR, canopy height data via https://portal.opentopography.org/datasets as open data and help improve local and global models of terrain.

Figure 18 Comparison of WorldDEM (5 m), FathomDEM and our GEDTM30 for two smaller areas in U.S. and Romania.

WorldDEM is a registered trademark by Airbus Defence and Space GmbH. Map data © Google Hybrid; Map data © OpenStreetMap.

Our results further indicate that many key terrain variables, especially hydrological variables, can be efficiently derived by overlapping the tiling rather than computing with whole continents or large watershed as in Amatulli et al. (2022). We show that the tiling can be optimized so that there are no significant artifacts, while computing is at the order of magnitude faster than if we would use global continents. In our case, we produced global mosaics of 15 key terrain variables at a high grid spacing of 30 m (land mask at 30 m consists of approximately 202 billion pixels in the EPSG:4326 projection system) using WhiteBox workflows for Python within days. We have also tested using SAGA GIS, GRASS GIS and GDAL for the variables derivation, and these could also be added to the computation workflow since all run also via command line.

Advantages of creating of localized global map using transfer learning

Our results have shown that transfer learning is a technique in machine learning in which knowledge learned from a task is re-used in order to boost performance on a related task. Research on transfer learning has shown that the merging of global and local data can benefit local prediction (Hauschild et al., 2022; Shen et al., 2022). In this work, our premise was that a single global model would not be able to adapt the complex environment on earth; hence we used the transfer learning to produce global maps with locally enhanced model predictions. Our GEDTM30 shows that it could potentially have a relatively similar accuracy to FathomDEM, but more modeling is needed to further increase the accuracy of GEDTM30.

The transfer learning technique can be further expanded to map other variables. The first obvious benefit is the great balance of local detail and global consistency. The pre-training models on a large and diversified source dataset leads to superior performance while transferring to a local fitting than only global or local modeling (Nie et al., 2022). The second benefit is to allow end users other than the original modeler to apply the pre-trained model to new local data, such as health or financial data (Gao et al., 2019; Hauschild et al., 2022; Segev et al., 2016). Nevertheless, we speculate that the transfer learning approach would not outperform global models universally, but that this should be tested case-by-case. In our case, modeling DTM from multisource of DSMs is straightforward as the base data, DSMs were consistent, and the sources of the training data for global and local models are the nearly same (e.g., global LiDAR data).

Regional discrepancy in terrain variables derivation processing times

Our results also show that the average time span of each deriving process provides a robust overview of the total computing time, easily scaling up to the total time number, factoring the time according to the grid spacing change (Fig. 14). However, this framework is not optimal if one tile (core) of the process is extremely slow and leads to an imbalance of computing time among cores. Therefore, we made a boxplot that records the computation time of each tile process as a single data point (Fig. 19). This helped us detect bottlenecks in computing (e.g., few BreachDepressionsLeastCost processes reach approximate 3,000 s), and the majority of the processes computed under 500 s per tile. Furthermore, Fig. 20 shows the top eight tiles (across Asia, Africa, and Europe) that were running BreachDepressionLeastCost for more than 1,200 s. We concluded that these tiles took excessive amount of computing time because they cover large water bodies, such as Lake Malawi for tile AF_E066N036T6, Targanyika lake, Victoria lake for tile AF_E060N024T6, and Capsian Sea for tile EU_E072N012T6, EU_E072N006T6, AS_E012N054T6, AS_E000N042T6, AF_E078N090T6. Thus, BreachDepressionLeastCost is set to run until it locates the lower cells. However, the large lakes in GEDTM30 are regarded as large flat areas, and the algorithm results in searching the whole lake, where in the case of the Capsian sea, it is approximately 10 times larger than, for example, Taiwan. Issues such as this illustrate that processing time can be significantly improved simply by carefully benchmarking derivation and understanding the behavior of the algorithms.

Figure 19 Boxplot of computing time for each process in 60 m grid spacing.

Figure 20 Equi7 tiles which take the longest process time for breach and fill depression.

Base map © Google Satellite.

We could not yet implement a method to reduce the discrepancy of tackling depression on a global scale. One could possibly focus on identifying the huge water bodies to prevent running the search on the non-land pixels. In addition, one could also test reducing the search distance for specific regions, which could then potentially further decrease the computing time. Also, we did not consider the use of hexagonal Discrete Global Grid Systems for storing DTM and variables (Li, McGrath & Stefanakis, 2022). The use of hexagonal DGGS appears to be definitively the most interesting way to further improve the precision and scalability of terrain variables derivation. We assume that our approach with tiling could be implemented in the hexagonal system, but we have not yet tested this.

Terrain uncertainty and environmental factors

We have further mapped uncertainty of GEDTM30, i.e., standard deviation of the predictive trees, and this matches the known technical issues with estimating height of terrain in the areas of complex terrain and covered with dense vegetation. Several other studies that cover the terrain height error, for example Bourgine & Baghdadi (2005) and Rizzoli et al. (2012), and also conclude that the DTM error increases along the slope and dense vegetated land cover, especially dense forests. These reported observations align with our uncertainty map.

One of the high uncertainty spots which has not been described so far are the water bodies. However, we postulate that it is not related to the characteristics of the land surface but rather to the covariates used in the prediction, especially from Landsat optical remote sensing: Landsat bands appear to have limited ability to represent inundations, wetlands, and similar. In addition, we discover some specific areas of high uncertainty in Greenland. This also probably comes from covariates since certain layers were not available for Greenland at all (e.g., neither canopy height models nor Landsat indices were available) (Lang et al., 2023; Potapov et al., 2020, 2021). The contribution of the terrain uncertainty map could be more than just looking at where the accuracy of the auxiliary data sources or the DTM needs to be improved to improve accuracy. We believe that the uncertainty map we produced (Fig. 13) can contribute to understanding the complexity of the terrain, offering spatial variability. For example, uncertainty could show the hot spots of elevation changes that correlate with susceptibility to terrain changes, such as landslides or earthquakes. The successive use cases require extra efforts to explore in the future.

Data limitation and next steps

GEDTM30 is a data fusion model that takes two main high quality sources of global digital surface models. GEDTM30 cannot perfectly exclude any artifacts. For example, GLO30 takes the Caspian Sea as a lake and sets the surface elevation as −29 m, whereas AW3D30 masked it as ocean and set it as 0 m. In addition, GLO30 has removed Azerbaijan from the map. Clear strips at the Azerbaijani border in the Caspian Sea are still visible in GEDTM30. In addition, our training points require further refinement, including additional cleaning and stratification using auxiliary data sources.

Another limiting issue of GEDTM30 is the accuracy in coastal areas. As Fig. 10 and Table 4 show, GEDTM30 and FABDEM suffer from similar accuracy issues in coastal areas. In these areas, GEDTM30 and similar products are barely useful for sea level rise studies and planning, such as analyses of future extreme water levels, which requires accuracy within 1 m (Dusseau, Zobel & Schwalm, 2023; Pronk et al., 2024). We try to resolve this issue by the balancing training points, because the coastal area is under sampled due to the portion of the land area. The near-coastline areas, namely the land within 100 km from the coast at an elevation of up to 100 m cover 9% of the global land area but carry almost 40% of the global population (Kummu et al., 2016; Reimann, Vafeidis & Honsel, 2023; Small & Nicholls, 2003), therefore it makes sense to expect DTMs with much higher accuracy in these areas.

One solution to further improve global DTM accuracy in the coastal area is to use a more simple data fusion of the best possible DTMs. For example, we previously experimentally merged ALOS, GLO30, MERIT DEM, and national DTMs by taking a 10% quantile of each DEM data set (Ho, Hengl & Parente, 2023). The result of a simple quantile 10% provided a promising DEM without artifacts at the source DEM edges, although the lack of 1 arc-s global land coverage DTM created inconsistency at certain locations. We further tested a merge of GEDTM30 and DeltaDTM by running the 10% quantile function pixel-wise on the valid pixels in the coastal area (Chincoteague) where the reference DTM is available. Figure 21 shows the comparison of the reference DTM, DeltaDTM, GEDTM30 and the 10% quantile merge DTM, which indicates a promising improvement compared to the reference DTM (the artifacts are barely visible in the transition area). Figure 21 also shows that the accuracy of the characteristics in coastal areas has improved significantly, especially in the forest cover and cropland areas. In addition, the DeltaDTM-derived hillshade has a greater difference than the reference, while the quantile function 10% propagates the GEDTM30 pattern but further removes objects from the original GEDTM30.

Figure 21 The 10% quantile ensemble of DeltaDTM and GEDTM30 in Chincoteague, the United States.

In summary, the simple data fusion approach shows the potential for a simple update by linear interpolation of multisource DTMs with GEDTM30 as the basis. Ultimately, the up-to-date topography service could be achieved to serve a wider community by running simple merges and updates. Even more interesting is data fusion by combining computer vision and multiscource data. The computer vision approach has been shown to tackle the height and removal mapping of objects more efficiently than if only point-based models were used (Lang et al., 2023; Uhe et al., 2025). The approach has the potential to remove objects from AW3D30 and GLO30 using a smaller number of covariates. However, we discovered that the underrepresentation of training data from developing countries has an impact on global accuracy (Fig. 12). Therefore, our next mission would be to try to improve DTM modeling by training the model on rasterized ICESat-2 and GEDI samples worldwide. This could, however, turn to be highly computational and might not be feasible at 1 arc s grid spacing.

In terrain variables derivation, the mixed-scale pyramid representation was applied for multiscale derivation. However, direct resampling by average is not optimal for geomorphometric analysis, and spatial filtering with progressive increase in window is preferable (Grohmann, 2015; Jenny, Jenny & Hurni, 2011; Kalbermatten et al., 2012; Newman, Lindsay & Cockburn, 2018; Newman et al., 2022). We did not implement direct progressive spatial filtering in production due to data storage issues. As the resolution is coarsened, the output variables are stored in a factor of 4 by each aggregation. Ultimately, 15 land variables at six levels take around 5.4 TB whereas processing in the finest resolution would take up to around 24 TB (after compression).

Recommend uses of data

GEDTM30 and its terrain variables cover a global land area excluding Antarctica. We are aware of artifacts on the surface of the Caspian Sea at the Azerbaijani border. It is recommended to mask out the Caspian Sea for further spatial analysis using GEDTM30. The water bodies such as lakes, pools, and coastal buffers are not applicable and should be masked out in the terrain variables. Multiscale global terrain variables have multiple uses in spatial predictive mapping, hydrology analysis, or even art purposes. The GEDTM30 and all derivatives produced are available under the CC-BY 4.0 license. The input data Copernicus DEM have been adapted. All rights are reserved ©DLR e.V. 2010–2014 and ©Airbus Defence and Space GmbH 2014–2018 provided under COPERNICUS by the European Union and ESA. Input data ALOS World 3D –30 m used for this article have been provided by AW3D of the Japan Aerospace Exploration Agency. The code used to generate GEDTM30 and optimize variable derivation is also available under MIT license through https://github.com/openlandmap/GEDTM30. We call on the research community to extend and improve these models under open data/open science principles to the benefit of everyone.

Conclusion

GEDTM30 is presented as a 1-arc-s global DTM based on machine-learning-based data fusion. We trained it using a global-to-local Random Forest model using ICESat-2 and GEDI (almost 30 billion of high quality points). Artificial and vegetation objects were removed from the merged surface of GLO30 and AW3D30, although some remaining vegetation artifacts are still visible. The model contains a consistent improvement in mapping terrain at built-up and dense tree-cover (tree cover 50%) area, with about 30% RMSE improved from Copernicus GLO30. In addition, an uncertainty map of terrain predictions was also produced, and which indicates that the uncertainty of terrain mapping is highest for areas with dense tree cover and steepness of terrain. We also performed a multiscale terrain variables derivation (at 30, 60, 120, 240, 480, and 960 m grid spacing), which can be extended in principle to an unlimited number of variables. Optimization in the tiling system, the land mask and the workflow design is fully documented in https://github.com/openlandmap/GEDTM30. However, a potential improvement in our workflow to further shorten the computing by tuning the breach and fill depression processes, which causes a significantly long time mainly trying to breach the depression in the huge inland water bodies.

GEDTM30 covers a decade of elevation data covering the time period 2006–2015 due to its underlying DEMs. Future research has to show how to repeat this exercise to complete the full space-time continuum for the next time slot of the years 2020 to 2030 to provide elevation differences to previous decades. We have also tested building up a simple update, ensemble DTM by simply taking quantile with other non-global coverage DTMs. The result shows that there is the possibility of a simple approach to create a seamless and easy-to-update open DTM service by combining other openly accessible DTMs. We envisage that this product will be operational (e.g., at the Copernicus Data Space Ecosystem or similar) and provided free of charge to the scientist, decision makers and businesses for their respective use cases. We propose two further development directions: (1) improve GEDTM30 by adapting to the computer vision approach, and (2) merge all local and regional compatible high-quality open source data available from OpenTopography.org to produce one single Ensemble DTM of the world.

This study builds upon multiple parallel initiatives within the Open-Earth-Monitor project and hence would not have been possible without kind contributions and work of many colleagues. We are especially grateful to Davide Consoli who helped produce long-term Landsat ARD, and Carmelo Bonannella for adjusting the research framework. Milutin Milenković and Johannes Heisig for helping filter the GEDI Level 2 data. Maarten Pronk from Deltares and Maria Hunter from Image Processing Laboratory and Geo (LAPIG) that helped derive individual photon information from ICESat-2 ATL08 version 6. The vast amount of data used in this article comes from the OpenTopography.org project, which is also our central target place for hosting these data. Finally, we also thank the Digital Elevation Model Intercomparison eXercise (DEMIX) Working Group that contributed knowledge in terms of terminology and comparison of digital elevation models.

Additional Information and Declarations

Competing Interests

Yu-Feng Ho, Leandro Parente, Martijn Witjes, and Tomislav Hengl are employed by OpenGeoHub. Hannes I. Reuter conducted this research in the capacity of a full-time employee of gisexperts. The research was carried out independently and is not related to, nor does it reflect the responsibilities, views, or interests of Eurostat or the European Commission.

Author Contributions

Yu-Feng Ho conceived and designed the experiments, performed the experiments, analyzed the data, prepared figures and/or tables, authored or reviewed drafts of the article, and approved the final draft.

Carlos H. Grohmann conceived and designed the experiments, analyzed the data, prepared figures and/or tables, authored or reviewed drafts of the article, and approved the final draft.

John Lindsay conceived and designed the experiments, performed the experiments, analyzed the data, authored or reviewed drafts of the article, and approved the final draft.

Hannes I. Reuter conceived and designed the experiments, analyzed the data, authored or reviewed drafts of the article, and approved the final draft.

Leandro Parente conceived and designed the experiments, performed the experiments, analyzed the data, authored or reviewed drafts of the article, and approved the final draft.

Martijn Witjes performed the experiments, authored or reviewed drafts of the article, and approved the final draft.

Tomislav Hengl conceived and designed the experiments, performed the experiments, prepared figures and/or tables, authored or reviewed drafts of the article, and approved the final draft.

Data Availability

The following information was supplied regarding data availability:

The Global-to-local modeling example is available at Zenodo: Yu-Feng Ho (2025). Global Ensemble Digital Terrain Model (GEDTM30): a Tile Example of Local Enhanced Modeling via Transfer Learning (1.0) [Data set]. Zenodo. https://zenodo.org/records/14914836

The GEDTM30 modeling and terrain variables derivation code and data are available at OpenLandMap GEDTM30 GitHub: https://github.com/openlandmap/GEDTM30 (MIT License).

The current version of code and data of GEDTM30 model and terrain variables are available in Zenodo: Yu-Feng Ho (2025). Global Ensemble Digital Terrain Model 30m (GEDTM30) [Data set]. Zenodo. DOI 10.5281/zenodo.14900180 (CC-BY 4.0 license). The code is written in the Python programming language and data are stored in Cloud Optimized GeoTIFFs.

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
