# Peer review of "GEDTM30: global ensemble digital terrain model at 30 m and derived multiscale terrain variables"

_PeerJ, doi:10.7717/peerj.19673_

## Round 0.1 · original submission · Minor Revisions

Dear Authors,

I have received two reports from the reviewers. Both consider your work to be of interest and provide suggestions on how to further improve its quality. In particular, they request a more detailed explanation of certain methodological aspects, clarification of the terminology, and suggest the potential inclusion or discussion of structural similarity metrics.

I invite the authors to carefully address the issues raised by the reviewers in the revision of the manuscript. I believe your article could represent an excellent contribution to PeerJ.

Reviewer 1 ·

Basic reporting

In recent years, various research groups have been working intensively to create new improved, compiled, and publicly available quasi-global terrestrial DEMs, such as NASADEM, MERIT DEM, Copernicus GLO-30 and GLO-90 DEMs, FABDEM, etc. One of the goals of creating compiled DEMs is to improve existing DEMs by filtering vegetation cover, noise, errors, and artifacts as well as filling voids. This goal is achieved by fusing heterogeneous data sources and applying various sophisticated data processing methods including machine leaning models. In this regard, the submitted manuscript is in line with this research trend.

The authors have a good knowledge of the specialized literature. The manuscript is well written and is well structured. On the whole, the manuscript makes a good impression. However, the authors use incorrect terminology, make a number of incorrect statements, and poorly describe two key data processing methods (see details below). The manuscript can only be published after these deficiencies are corrected.


1. Describing DEMs, the authors systematically, throughout the article, incorrectly use the term ‘resolution’ instead of ‘grid spacing’. For example: “1 arc second resolution”, “6 standard resolution (30, 60, 120, 240, 480 and 960 m)”, “30 m resolution”, “spatial resolution of 30 m”, “5 arc-degree resolution”, “15-arc-second resolution”, and so on.

For difference between spatial resolution of gridded data and grid spacing, see p. 5 in [Guth et al., 2021. Digital Elevation Models: Terminology and Definitions. Remote Sensing., 13(18), 3581]. In all such cases, the term 'resolution' should be replaced by 'grid spacing'.


2. Describing derivation of land surface attributes from DEMs, the authors systematically, throughout the article, incorrectly use the term ‘parameter’ in relation to slope, various types of curvatures, catchment area, topographic wetness index, etc. For example: '15 standard land surface parameters', 'multiscale land surface parameters', 'deriving land surface parameters', 'basic geomorphometric and hydrological DTM parameters', etc.

Hillshade, slope, openness, several curvatures, shape index, topographic position index, specific catchment area, slope length, topographic wetness index are NOT parameters of topography or land surface. These are functions. In particular, all curvatures and shape index are functions of the first and second partial derivatives of elevation. In geomorphometric literature, these functions are called morphometric / topographic variables or attributes. These functions cannot be parameters by any definition of the term ‘parameter’. In the manuscript, the term ‘parameter’ must be replaced by ‘variable’ or ‘attribute’.


3. To describe derivation of land surface attributes from DEMs, the authors systematically, throughout the article, incorrectly use the term ‘DTM parametrization’.

Parametrization is the process of defining or choosing parameters. You do NOT define or choose parameters. You derive / compute / calculate morphometric attributes. The expression ‘DTM parametrization’ should be replaced by, for example, ‘geomorphometric derivations’ or ‘geomorphometric calculations’.


4. In line 409, the authors stated: “There are two types of land surface parameters: regional and local (Olaya, 2009b).”

This was established long before Olaya. See [Shary, P.A. (1995). Land surface in gravity points classification by a complete system of curvatures. Mathematical geology, 27, 373–390.]


5. In lines 127-128, the authors stated: “In addition to modeling DTM, the extraction of measures and spatial features from DEMs (geomorphometry) is an emerging field.”

This is absolutely not true. Derivation of morphometric attributes from DEMs (e.g., slope, various types of curvatures, catchment area, etc.) has been usual geoinformatic procedures since at least the early 1990s, although the first algorithms for calculating morphometric characteristics were developed in the late 1970s and 1980s.

Experimental design

6. Pages 15-16: What a specific method was used to calculate maximal curvature, minimal curvature, profile curvature, tangential curvature, ring curvature, and shape index? You indicated that this method uses ‘a polynomial fit of the elevations within the 5×5 neighborhood surrounding each cell’. The reader is not interested in what specific software function you used for the calculations. The reader is interested in what specific method was used. Please provide a reference to the method.

7. It is unclear from the manuscript whether the DEM reprojection involved a transition from the spheroidal equal angular grid with the 1 arc second grid spacing to the plane square grid with the 30-m grid spacing. Please, clarify this point.

Validity of the findings

'no comment'

Additional comments

'no comment'

Reviewer 2 ·

Basic reporting

The manuscript is generally well-written, employing clear and professional English suitable for scientific publication. The language used is largely unambiguous and technically sound, adhering to professional standards of expression. The authors have made a commendable effort to define terminology, particularly distinguishing between DSMs and DTMs, which is crucial given the focus of the work. The text flows logically, guiding the reader through the complex methodology and results.
The introduction provides sufficient background and context, effectively situating the research within the broader field of global elevation modeling and geomorphometry. It adequately reviews the history and limitations of existing global DEMs (SRTM, ASTER, AW3D30, CopernicusDEM) and recent DTM efforts (MERIT, FABDEM, CoastalDEM, FathomDEM), clearly identifying the knowledge gap: the need for a fully open, globally consistent, high-resolution DTM derived using modern data sources and methods, alongside efficient parameterization techniques. Relevant prior literature is appropriately referenced throughout the introduction and discussion sections, supporting the claims and methodologies presented.
The article follows a standard professional structure (Abstract, Introduction, Materials and Methods, Results, Discussion, Conclusion, Declarations, References), which aids readability. Figures and tables are numerous and generally relevant to the content. Figures like the workflow diagram (Fig. 1), the global-to-local modeling explanation (Fig. 2), the multiscale parameterization illustration (Fig. 3), and the tiling system map (Fig. 4) are particularly helpful in visualizing the complex processes involved. The visual comparisons of GEDTM30 with source DSMs (Fig. 7, 8) and other DTMs (Fig. 9, 10, 11, 17) are effective in demonstrating the model's performance and limitations. Tables clearly summarize input covariates (Table 1), accuracy metrics (Table 3, 4, 5), and parameter details (Table 2). The resolution of figures appears adequate in the provided format.
The authors explicitly state that the GEDTM30 dataset, derived parameters, and the code used for generation and parametrization are publicly available via Zenodo and GitHub under open licenses (CC-BY 4.0 and MIT, respectively). This commitment to open data and reproducibility aligns strongly with the journal's data sharing policy and is a significant strength of the work.
The submission represents a self-contained and appropriate unit of publication. It describes the complete process from data acquisition and preparation through model development, validation, parameter derivation, and optimization. The results presented directly address the research objectives outlined. There is no indication that a coherent body of work has been inappropriately subdivided.

Experimental design

The research presented is original primary work focused on developing a novel global DTM and associated parameters. This falls well within the typical aims and scope of journals covering remote sensing, geoinformatics, Earth observation, and geomorphometry.
The research question is well-defined: how to create an improved, globally consistent, open-access 30m DTM by fusing multiple state-of-the-art datasets (DSMs, lidar, ancillary layers) using machine learning, and how to efficiently derive multiscale parameters from it. The paper clearly articulates the knowledge gap this addresses, highlighting limitations in existing open global DTMs regarding accuracy (especially object removal), consistency, full openness, and the computational challenges of generating high-resolution parameters globally. The contribution is stated as providing both the improved DTM (GEDTM30) and an optimized, reproducible workflow for its creation and parametrization.
The investigation appears to have been conducted rigorously and to a high technical standard. The use of multiple state-of-the-art input datasets (GLO30, AW3D30, ICESat-2, GEDI, canopy/building height models, Landsat ARD) is appropriate. The data preparation steps, including extensive cleaning, filtering, and stratification of the massive lidar training dataset (over 30 billion initial points reduced to millions for training/tuning), demonstrate considerable effort to ensure data quality and representative sampling. The application of a global-to-local transfer learning approach with Random Forests is a sophisticated method to balance global consistency with local accuracy. The derivation of multiscale parameters uses established geomorphometric techniques implemented in recognized software (Whitebox Workflows). The optimization of the parametrization workflow using an extended Equi7 tiling system addresses significant computational challenges. The research appears to conform to ethical standards, primarily using publicly available or open datasets and providing code and outputs openly.
The methods are described with substantial detail, likely sufficient for replication by investigators with appropriate expertise and computational resources. The authors meticulously document data sources (including versions and temporal ranges), preprocessing steps (gap-filling, noise removal, lidar filtering criteria, stratification logic), the modeling framework (Random Forest, global-to-local transfer learning, hyperparameter tuning approach, prediction generation), post-processing steps (filtering), validation datasets and metrics, the specific land surface parameters calculated and the Whitebox Workflows functions used, and the tiling/parallelization strategy (Equi7 extension, overlap distances, land mask creation). The provision of the actual code further enhances reproducibility.

Validity of the findings

The study appropriately adheres to the journal's policy of not assessing subjective impact or novelty. It focuses on presenting the methodology and validation of the GEDTM30 product. While it is not strictly a replication study, the approach builds upon and synthesizes established techniques (data fusion, machine learning for object removal, geomorphometric analysis) and compares its results extensively with prior state-of-the-art global DTMs, providing value to the literature.
The underlying data used for model training (lidar points) and the final DTM product are stated to be provided openly. The validation framework appears robust. The authors use independent, high-quality airborne lidar DTMs from various locations (though acknowledging geographic bias towards Europe/Americas) and globally distributed GNSS station data as reference datasets. Comparisons are made not only between GEDTM30 and the reference data but also include the source DSMs (GLO30, AW3D30) and other prominent global DTMs (MERIT, FABDEM, FathomDEM), allowing for a comprehensive assessment of relative performance. Standard statistical metrics (RMSE, MAE, ME, Standard Deviation) are used appropriately to quantify errors. The analysis considers performance across different land cover types (built-up, tree cover densities, cropland etc.) and terrain settings (mountainous, coastal plain, boreal forest), providing insights into the model's strengths and weaknesses in various environments. The generation of an uncertainty map (based on Random Forest prediction standard deviation) adds further value by indicating areas of lower model confidence. While FathomDEM shows slightly better performance in some validation tests against reference DTMs, GEDTM30 performs competitively, particularly against GNSS data, and demonstrates clear improvement over the source DSMs in object removal.
The conclusions drawn are generally well-stated and linked back to the original research questions. The authors summarize the creation of GEDTM30, the use of the global-to-local framework, the validation results (including improvements over source DSMs and comparisons with other DTMs), the generation of the uncertainty map, and the successful implementation of the multiscale parametrization workflow. Importantly, the conclusions are appropriately limited to those supported by the results. The authors acknowledge limitations, such as remaining artifacts, the slightly lower accuracy compared to FathomDEM in certain tests, specific issues in coastal areas, potential refinements needed for training data, and bottlenecks in the parametrization workflow (e.g., large water bodies). They do not overstate the performance or make unsupported claims. The discussion also appropriately contextualizes the findings and suggests avenues for future work, such as incorporating computer vision or fusing with national DTMs.

Additional comments

The present review, after initial drafting, has been AI-assisted/edited.

This manuscript presents a significant contribution to the field of global terrain modeling. The development of GEDTM30 as a fully open-source, 30m resolution global DTM, derived using a robust data fusion and machine learning approach, is highly valuable. The integration of massive ICESat-2 and GEDI lidar datasets for training is a key strength. The global-to-local transfer learning framework is an innovative method to tackle the challenge of creating a model that performs well across diverse global landscapes. Furthermore, the successful implementation and optimization of the workflow for generating a comprehensive suite of 15 multiscale (30m-960m) land surface parameters is a major achievement, addressing significant computational hurdles. The commitment to open data and open code is exemplary and greatly enhances the potential impact and utility of this work for the research community.
The validation is thorough, comparing GEDTM30 against multiple independent reference datasets and state-of-the-art competing global DTMs. The results honestly portray both the strengths (significant improvement over source DSMs, competitive performance) and weaknesses (slightly lower accuracy than FathomDEM in some tests, coastal zone issues, remaining artifacts) of the developed product. The discussion thoughtfully addresses limitations and proposes concrete next steps.

POTENTIAL IMPROVEMENT
One area for potential enhancement relates to the validation metrics used. While RMSE, MAE, and ME provide valuable information about point-wise vertical accuracy and bias, they do not fully capture the structural or geomorphological fidelity of the terrain representation. Errors in DTMs are often spatially correlated, and metrics like RMSE can be heavily influenced by outliers without reflecting how well overall landscape features (e.g., ridges, valleys, slopes) are preserved. The authors partially address this through visual inspection using hillshades and cross-sections, which is insightful. However, this evaluation could be quantitatively supported by incorporating a metric that considers spatial structure to further strengthen the validation. The Structural Similarity Index Measure (SSIM), commonly used in image quality assessment, could be adapted for comparing DTMs (it has already been used also for DTM interpolation/inpainting purposes). SSIM evaluates similarity based on luminance, contrast, and structure, potentially offering a more nuanced assessment of how well the modeled terrain structurally resembles the reference DTMs, complementing the error magnitude information from RMSE and MAE. Suggesting or exploring the use of SSIM (or similar structure-aware metrics) in future validation efforts or as an additional metric in this study could provide deeper insights into the quality of the terrain representation beyond simple vertical offsets. SSIM goes exactly in the direction of perceived similarity, considering the structure (relative patterns) and being robust against noise or diversity in spatial arrangement that could severely impact ME-based analysis usually providing “good” results also when it should not be the case. This could be particularly relevant when comparing DTMs generated using different methods (e.g., machine learning fusion vs. filtering vs. computer vision) which might exhibit different types of artifacts or smoothing effects.
As a side note, in case SSIM metrics are integrated, I suggest Matlab (or Matlab-like) implementation (window size 11, sigma 1.5, k1 0.01, k2 0.03).
Minor points:
• The paper mentions filtering speckle noise using adaptive and bilateral filters only on tree and building cover pixels. A brief (maybe obvious) justification for applying filters only to these areas (presumably where object removal introduces potential artifacts) versus globally might be useful.
• The discussion on processing time bottlenecks for BreachDepressionsLeastCost over large water bodies is interesting. While solutions are proposed for future work, confirming that the current parameters (e.g., max search distance 100 pixels ) did not unduly compromise hydrological parameter quality in other complex (but non-lake) terrains might be reassuring.
Overall, this is a strong paper presenting a valuable dataset and methodology. The research is well-executed, thoroughly documented, and openly shared.

---

## Round 0.2 · accepted · Accept

Dear authors, the new version of your manuscript has been revised for the second time by previous reviewers who both appreciated the strong effort you put to addreess all the raised issues. The manuscript is thus now ready to be published. I'm convinced this study is an excellent contribution that will be be certainly of great interest for the PeerJ reader.

A small side note: when revising the proofs of the article please carefully check references because some information is missing (e.g line 1006 Florinsky et al. (2017) the correct reference is Florinsky, I. V. (2017). An illustrated introduction to general geomorphometry. Progress in Physical Geography, 41(6), 723–752. doi: 10.1177/0309133317733667)

Reviewer 1 ·

Basic reporting

The authors have worked hard to improve their article. All my comments were scrupulously taken into account. I believe that the article can be published.

Experimental design

no comment

Validity of the findings

no comment

Additional comments

no comment

Reviewer 2 ·

Basic reporting

The manuscript has seen improvements in clarity and terminology. The authors have largely addressed the points raised in the previous review.

Experimental design

The authors have provided more detailed explanations for certain methodological aspects, which enhances the understanding of the experimental design.

Validity of the findings

The changes made contribute to strengthening the validity of the findings. The discussions on the consistency checks and the limitations are well-received.